# Pregnancy and Childbirth Fear of Women from Poland, Greece, Turkey, Belarus, and Russia

**DOI:** 10.3390/jcm13133681

**Published:** 2024-06-24

**Authors:** Agnieszka Kułak Bejda, Lambrini Kourkouta, Areti Tsaloglidou, Konstantinos Koukourikos, Ilknur Aydin Avci, Dilek Çelik Eren, Andrei Shpakou, Natallia Khvoryk, Liudmila Hutsikava, Napoleon Waszkiewicz

**Affiliations:** 1Department of Psychiatry, Medical University of Bialystok, 15-089 Bialystok, Poland; napoleonwas@yahoo.com; 2Nursing Department, International Hellenic University, 57400 Thessaloniki, Greece; laku1964@yahoo.gr (L.K.); aretitsa2010@hotmail.com (A.T.); kokaea@yahoo.gr (K.K.); 3Department of Nursing, Faculty of Health Sciences, Ondokuz Mayis University, Samsun 55139, Turkey; ilknursezera@hotmail.com (I.A.A.); dilek.celik@omu.edu.tr (D.Ç.E.); 4Department of Integrated Medical Care, Faculty of Health Sciences, Medical University of Bialystok, 15-089 Białystok, Poland; andrei.shpakou@umb.edu.pl; 5Department of Obstetrics and Gynecology, Grodno State Medical University, 230009 Grodno, Belarus; kafedra2.bsmp@mail.ru (N.K.); klam4@mail.ru (L.H.)

**Keywords:** pregnancy, postpartum, childbirth, fear

## Abstract

**Background/Objectives**: Pregnancy and childbirth in many women cause various situations, from physical to emotional. The analysis of selected factors affecting pregnancy and childbirth in a group of women from Poland, Greece, Turkey, Belarus, and Russia and self-assessment of their impact on fear of pregnancy and childbirth. **Material and Methods**: A total of 2017 women were surveyed, including 584 pregnant women, 528 postpartum women, and 906 non-pregnant and never-pregnant women. The study used an original questionnaire, including questions on socio-demographics and the history of pregnancy and childbirth. The material status of the respondents was assessed using the Family Affluence Scale. **Results**: The age range of respondents was 19–50. The surveyed women most often reported (*p* < 0.001) fear (n = 928) and excitement (n = 901). A positive correlation was found between anemia, infections, and fear in women from Belarus. No correlation was found between low, average, or high material status and the perception of pregnancy in women from the studied countries. Significantly (*p* < 0.001), women from Turkey had more children (*p* < 0.001) than women from other countries. With the exception of Russia, statistical correlations were shown between the feeling of fear of pregnancy and childbirth and the way the previous/current pregnancy was delivered and the experience of an artificial/natural miscarriage in the past. In general, 630 women had given birth vaginally, and 283 women had given birth by cesarean section. In the group of currently pregnant women, 22 women had had natural miscarriages in Belarus, 37 in Poland, 27 in Greece, 29 in Turkey, and 9 in Russia. **Conclusions**: When thinking about pregnancy, respondents in all groups most often felt joy and excitement, as well as fear and excitement, about childbirth. Natural miscarriage was experienced most frequently by women in Poland and Greece and induced in Belarus. The largest number of women who had cesarean section were from Poland and Belarus.

## 1. Introduction

Motherhood is recognized as one of a woman’s most significant roles, often regarded as the central and dominant element of womanhood. It is a time of discovering one’s capabilities, strengthening perseverance and efficacy, learning a new role, and assuming more responsibilities.

During pregnancy, women may experience various situations, from physical to emotional, which affect both the mother’s and baby’s health [1]. Thus, needs and concerns arise related to their bodily sensations, changes in lifestyle, and fears about the possibility of miscarriage [2].

The timing of childbirth causes many concerns. The mother’s knowledge and expectations often influence childbirth. One of the most common feelings reported by pregnant women concerning childbirth is fear. This increases the perception of childbirth as a traumatic and painful process [3]. Many women demand a cesarean section delivery, believing it to be safer and more comfortable for them and the baby [4].

There is also a noticeable upward trend in the rate of cesarean sections worldwide, with an average increase of 9% between 2000 and 2015 [5]. On individual continents, the changes are distributed differently: in Western Europe, it was 7.3%; in Eastern Europe and Central Asia, 15.4%; in Northern America, 7.7%; and in the Middle East and Northern Africa, as much as 10.6%. In Poland, the rate of cesarean sections is one of the highest in Europe, at 42.2% [4].

In recent years, Poland has seen a change in fertility patterns, a decline in the number of births in all reproductive age groups, and a decrease in the absolute number of births [6]. The fertility rate for Poland in 2023 was 1.467 births per woman, a 0.62% increase from 2022. The fertility rate for Belarus in 2023 was 1.736 births per woman, a 0.29% increase from 2022. The fertility rate for Russia in 2023 was 1.825 births per woman, a 0.05% increase from 2022. The birth rate for Greece in 2023 was 6.930 births per 1000 women, a 2.56% decline from 2022. The fertility rate for Turkey in 2023 was 1.994 births per woman, a 0.85% decline from 2022. In 2021, the rate was 1.33, and was thus below the EU average and well below the replacement level [7].

Several factors influence birth rates, impacting population growth and demographics [8,9,10,11]. Generally, higher levels of wealth and education are associated with decreased fertility rates. Educated individuals often delay childbirth and have fewer children. As more women participate in the workforce, they tend to have fewer children due to career demands and lifestyle choices. Urban areas often exhibit lower birth rates compared to rural regions. Urbanization leads to smaller families and different lifestyle priorities. It is well known that increased maternal age is linked to lower fertility. First-time mothers today tend to be older, contributing to declining birth rates. Birth rates are dynamic and influenced by a complex interplay of social, economic, and cultural factors.

Also, fear of childbirth may affect the birth rate. Fear of childbirth is a natural reaction for many expectant mothers. Women often experience uncertainty about the labor process. They often fear that something might go wrong or be painful. Pregnant women are increasingly exposed to the view that pregnancy and childbirth are intrinsically dangerous, requiring medical monitoring and management. Societal pressures that dictate appropriate behaviors during pregnancy are applied to women. These changes have resulted in an increased perception of risk for pregnant women [6,8].

It is essential to learn about the birthing process and attend childbirth preparation classes [12]. Many women worry about significant weight gain during pregnancy [13]. Women are often afraid that they will be less attractive to their partner after giving birth. Some women worry that there may be complications with the baby. Fear of pregnancy is known as tokophobia [14]. It is a specific anxiety disorder where individuals experience an irrational and excessive fear related to pregnancy and childbirth. Women with this phobia may avoid becoming pregnant altogether or opt for a cesarean section to avoid vaginal birth. Some women may experience sleep disturbances, panic attacks, and nightmares.

Numerous studies have been published on the fear of childbirth having a significant impact on delivery outcomes, resulting in a dramatic increase in cesarean births [6,8,14,15,16,17,18]. 

It is suggested that some women are more susceptible to fear of childbirth than others. The etiology of tokophobia is multifactorial and can be associated with different factors such as susceptibility to anxiety or depression [15] and other psychiatric disorders, well-being in interpersonal relationships [16], and traumatic experiences of former births [17]. Some women described themselves as lonely or with low self-esteem [18]. The need for psychiatric care and the presence of traumatic stress symptoms are reported outcomes together with prolonged labor, use of an epidural, and obstetric complications.

To our knowledge, similar studies comparing factors affecting pregnancy and childbirth in women from Poland, Greece, Turkey, Belarus, and Russia are unavailable.

No similar studies have been conducted in Belarus or Russia. Several studies are from Poland [19,20,21]. Only a few studies from Turkey and Greece have assessed fear of childbirth and pregnancy. A Turkish study from 2020 found that 82.6% of the women had a fear of childbirth [22]. Economic status, parity, previous birth experience, and preferred delivery method affected the fear of childbirth among women. In a Greek study [23], a Validation of the Childbirth Attitudes Questionnaire was used, and 145 late-pregnancy women participated. Eighty percent of women reported that they would intend to have a normal vaginal delivery, while 20% of participants stated that they preferred cesarean section.

In this study, we tried to compare factors that could affect pregnancy and childbirth in women from Poland, Greece, Turkey, Belarus, and Russia.

### Aim of the Study

The study’s main aim was to analyze factors that may affect the perception of pregnancy and childbirth in a group of pregnant and postpartum women from Poland, Greece, Turkey, Belarus, and Russia.

## 2. Material and Methods

The study was conducted with approval from the Bioethics Committee APK.002.587.2021, Białystok, Poland. All participants provided written informed consent for participation. Participation was voluntary and anonymous. Participants could withdraw from the study at any point without feeling obligated to continue. Personally identifiable data were not collected.

The sample selection was purposive. A cross-sectional study was carried out amongst the 2017 women surveyed, including pregnant and postpartum women who were not pregnant and had never given birth from Poland, Greece, Turkey, Belarus, and Russia. The sample selection was purposive. The pregnant women were recruited from Obstetrics Departments, the postpartum women were recruited from General Practice Clinics, and the group of women who had never given birth and were not pregnant included female students and university workers in the studied countries. Surveys were collected between November 2021 and December 2022 during the COVID-19 pandemic. This could have affected the collection of the surveys. The questionnaires were given to the participants by the authors of the study. All completed questionnaires from Greece, Turkey, Belarus, and Russia were sent to Poland. A statistician analyzed the data from the questionnaires.

The inclusion criteria were as follows: pregnant and postpartum women over 18 years of age who could communicate in the mother language of the country where they were contacted were included in the study.

The exclusion criteria were as follows: women were excluded if they were under 18 years of age and could not communicate in the mother language of the country where they were contacted.

### 2.1. Measures

The study used a diagnostic survey method involving an original questionnaire about age, marital status, place of residence, education, material status, and associations of respondents regarding pregnancy and childbirth (joy, excitement, fatigue, helplessness, fear, hospital, house, and loneliness) with and health problems in the last year. In the group of pregnant and postpartum women, the respondents were additionally asked about the number of children, spontaneous and induced abortions, duration of the delivery, total number of deliveries, type of the last delivery, and participation in childbirth classes.

The material status of the respondents was assessed using the Family Affluence Scale (FAS) [24]. The FAS was developed within the ‘Health Behaviour in School-Aged Children: WHO Collaborative Cross-National Study’ (HBSC), which aims to collect comparative data on adolescent health and its determinants. The scale is a socioeconomic proxy for family wealth in studies where reliable, objective information about family wealth is impossible to obtain. This questionnaire includes the following questions: Does your family own a car, van, or truck? (Scores 1—No, 2—Yes one, and 3—Yes two or more). Do you have a bedroom for yourself? (Scores 1—No, 2—Yes). How many computers do your family own (including laptops and tablets, not including game consoles and smartphones)? (Scores 1—None, 2—One, 3—Two, and 4—Two or more). How many bathrooms (room with a bath/shower or both) are in your home? (Scores 1—None, 2—One, 3—Two, 4—More than two). Does your family have a dishwasher at home? (Scores 1—No, 2—Yes). How often did you and your family travel out of [insert country here] for a holiday/vacation last year? (Scores 1—Not at all, 2—Once, 3—Twice, and 4—More than twice). A simple sum score is the starting point for computing relevant socioeconomic indices. The Cronbach’s α of the scale is 0.643.

### 2.2. Statistical Analyses

The statistical analysis was performed based on the Statistica 13.0 PL software. The Shapiro–Wilk test was used for the test of normality. A *t*-test was used to compare the means of the two groups. The chi-square test was used for sample sizes of five and larger. This test was used to compare the percentages between countries. The correlations between variables were calculated using Spearman’s rank correlation analysis. We used the interpretations of the r values: poor correlation is r = 0.1–0.2; fair is r = 0.3–0.5; moderate is r = 0.6–0.7; and very strong is r = 0.8–0.9. Statistically significant differences were defined at *p* < 0.05.

### 2.3. Results

The study involved 2017 women, including 584 pregnant women, 528 postpartum women, and 906 women who were not pregnant and had never given birth from Poland, Greece, Turkey, Belarus, and Russia. Married women dominated all study groups. The mean age of 19.2 ±19 years for women who had never given birth in Belarus was significantly (*p* < 0.001) lower compared to Poland (33.5 ± 33) and Turkey (28.3 ± 26). In the pregnant group, it was 29.2 ± 29 years in Belarus, 30.4 ± 30 in Poland, 34.7 ± 33 in Greece, 27.7 ± 27 in Turkey, and 27.7 ± 29 in Russia. In the postpartum group, it was 28.7 ± 29 years in Belarus, 31.0 ± 30 in Poland, 37.6 ± 36 in Greece, 28.1 ± 28 in Turkey, and 28.1 ± 28 in Russia. All groups and countries were significantly (*p* < 0.001) dominated by urban residents, with an overall number of 1699 vs. 318. In Poland, respondents with a master’s degree significantly (*p* < 0.001) dominated (n = 201) compared to other countries. Statistically, the largest number of students were from Russia, n = 262, compared to other countries (Table 1).

## 3. Results 

Assessment of material status 1 with the FAS allowed us to conclude that the average material status was high (above 5 points) in all groups and countries (Table 1). The largest number of women with a very low material status was in Belarus, with a low and average material status in Turkey, and a high material status in Poland. However, in the study group, women with a high material status dominated with 72%.

No correlation was found between low, average, or high material status in the FAS and the perception of pregnancy in women from Belarus. Similarly, no correlation was found between low, average, or high material status in the FAS and the perception of pregnancy in women from Poland, Greece, Turkey, and Russia.

Respondents were asked what associations they had with regard to pregnancy. Among the women who had never been pregnant, those from Belarus and Russia most often (*p* < 0.001) reported excitement (n = 43 and n = 80, respectively), while in Poland and Greece they most often mentioned joy (n = 108 and n = 96, respectively). Pregnant women from Belarus, Poland, and Greece reported joy more often (n = 130, n = 145, and n = 102) than others. Details are shown in Table 2.

Respondents were asked to list the diseases affecting them during the survey. In the group of pregnant and postpartum women from Poland, the most common diseases were infections, diabetes, and anemia. In women from Belarus, they were infections and anemia. A positive correlation was found between anemia, infections, and fear in women from Belarus. Details of the results are shown in Table 3.

Respondents were also asked what associations they had regarding childbirth. For women who had never been pregnant, those from Greece (n = 84) and Russia (n = 72) mentioned fear significantly (*p* < 0.001) more often than other women. Pregnant women from Belarus (n = 102) and Turkey (n = 78) reported excited significantly more frequently. A total of 64.2% of pregnant women reported childbirth fear. Postpartum women from Belarus (n = 63) and Turkey (n = 67) reported excitement. In general, women from Belarus (n = 199) and Turkey (n = 234) more often (*p* < 0.001) reported that childbirth evoked excitement. Generally, all women from the five surveyed countries reported fear (n = 928) and excitement (n = 901) most often (*p* < 0.001). Details are shown in Table 4.

Pregnant women at the time of the survey had an average of 1.4 ± 0.7 children in Belarus, 1.4 ± 1.1 in Poland, 2.0 ± 0.9 in Greece, 3.1 ± 1.4 in Turkey, and 1.1 ± 1.2 in Russia. Significantly, women from Turkey had more children (*p* < 0.001) than others. Details are shown in Table 5.

In the group of pregnant women, only 63 (25.2%) from Belarus, 37 (12%) from Poland, 8 (3.5%) from Greece, 26 (12.4%) from Turkey, and 4 (4.2%) from Russia participated in childbirth classes. In the group of postpartum women, 1 (0.4%) from Belarus, 59 (19.2%) from Poland, 6 (2.6%) from Greece, 12 (12.6%) from Russia, and none from Turkey participated in childbirth classes. Significantly (*p* < 0.001), the highest number of pregnant and postpartum women, n = 867 (79.3%), did not attend childbirth classes compared to those attending, n = 216 (19.8%).

Significantly (*p* < 0.001), the highest number of births were reported both by pregnant and postpartum women from Poland and the lowest from women from Russia (Table 6).

Respondents who had given birth previously or were in the postpartum period were asked to specify the type of delivery. The highest percentage, 29 (30.5%) natural births, were reported by women from Russia, and the lowest was 63 (20.4%) from Poland. Significantly (*p* = 0.002), women from Greece reported 72 (31.3%) cesarean sections compared to 10 (10.5%) from Russia. In total, 283 women gave birth by cesarean section, a few more than the 264 women who gave birth naturally. Details are shown in Table 7.

Postpartum respondents were asked to report how long their delivery lasted. The duration of childbirth was significantly (*p* < 0.001) the longest in Greece compared to other countries. In Belarus, it lasted 5.8 ± 3.8 h; in Poland, 5.6 ± 3.6; in Greece, 10.8 ± 8.9; in Turkey, 4.6 ± 4.3; and in Russia, 5.3 ± 4.2 h.

The women surveyed were also asked whether they had experienced a spontaneous abortion. In the group of pregnant women, 22 had had spontaneous abortions in Belarus, 37 in Poland, 27 in Greece, 29 in Turkey, and 9 in Russia. There were no induced abortions in the group of subjects from Turkey. Induced abortions occurred in 12 women in Belarus, 11 in Greece, 6 in Poland, and 2 in Russia. The results are shown in Table 8.

We also analyzed relationships between the respondents’ associations regarding pregnancy and the method of an earlier delivery.

In the Polish group, among women who were currently pregnant and had previously given birth naturally, significant relationships were found with the feeling of excitement (R = 0.254; *p* = 0.008), anxiety (R = −0.207; *p* = 0.032), and the feeling of loneliness (R = −0.251; *p* = 0.026), and in women currently intending to give birth by cesarean section with a sense of helplessness (R = −0.193; *p* = 0.046).

In the group from Belarus, significant correlations were found only in terms of perceived loss, both in the group of pregnant women who had previously given birth vaginally (R = −0.441; *p* = 0.0002) and those currently after vaginal delivery (R = −0.451; *p* = 0.0001).

In the Greek group, a significantly negative relationship was found in the area of fear felt in the group of women currently after natural childbirth (R = −0.377; *p* = 0.000).

In the Turkish group, significant negative relationships were found in terms of perceived fear in pregnant women after induced labor (R = −0.706; *p* = 0.000) or cesarean section (R = −0.312; *p* = 0.039). No significant correlations were found among Russian women. The results are shown in Table 9.

Next, the respondents’ associations regarding pregnancy were analyzed between women who had and who had not experienced a natural miscarriage in the past.

In the group from Poland, among currently pregnant women, significant correlations were found between the feeling of anxiety (R = −0.246; *p* = 0.001) and the feeling of loneliness (R = −0.310; *p* < 0.001); in the group from Greece, currently pregnant women experienced anxiety (R = −0.223; *p* = 0.017); in the Russian group, they experienced excitement (R = 0.388; *p* = 0.007); and women from Belarus experienced anxiety (R = −0.572; *p* < 0.001), a sense of helplessness (R = −0.571; *p* < 0.001), and loneliness (R = −0.396; *p* < 0.001).

In the group of currently postpartum women from Belarus, significant negative correlations were also found for fear (R = −0.266; *p* = 0.001) and sense of helplessness (R = −0.492; *p* < 0.001). The results are shown in Table 10.

The respondents’ associations regarding childbirth were also analyzed between women who had and who had not experienced an artificial miscarriage in the past.

In the group of pregnant women, significant differences were found between miscarriage in the group from Belarus and the feeling of joy (R = 0.228; *p* = 0.02) and fatigue (R = −0.341; *p* < 0.001) in the face of fear (R = −0.278; *p* = 0.004) and loss (R = −0.197; *p* = 0.044), and in the Russian group with fear (R = −0.551; *p* < 0.00).

Additionally, in the group of women from Greece who were currently pregnant, significant differences were found between a previous artificial miscarriage and excitement (R = 0.205; *p* = 0.035) and fatigue (R = −0.286; *p* = 0.003). Details are shown in Table 11.

The respondents’ associations regarding childbirth and the method of earlier delivery were also analyzed. Significant differences were only found in the group of women from Poland, including those who were currently pregnant and had given birth vaginally, with excitement (R = 0.254; *p* = 0.008) and joy (R = −0.207; *p* = 0.032) and those who had given birth by cesarean section (R = 0.222; *p* = 0.021). Women who had just given birth had a feeling of helplessness (R = 0.193; *p* = 0.046). The results are shown in Table 12.

## 4. Discussion

Our study was performed during the COVID-19 pandemic, a difficult time for all people, particularly pregnant women. The COVID-19 epidemic led to increased fear, stress, and anxiety among pregnant women. They often perceived childbirth during this time as a threat to their well-being and health [20,25,26]. A study funded by the National Institutes of Health found that women who gave birth during the pandemic, especially in communities with COVID-19 outbreaks, were more likely to experience traumatic childbirth. This included symptoms of intense anxiety or post-traumatic stress disorder triggered by the birthing experience [27]. Pregnant women were very often worried about the virus’s impact on their pregnancy and unborn babies. These negative factors could also impact our results. 

Our findings are in agreement with previous reports on childbirth fear. In the present study, 64.2% of pregnant women reported childbirth fear. In a study from Slovenia, 75% of pregnant women reported low to moderate tokophobia, whereas 25% exhibited high or very high fear of childbirth [28]. A European study of 6870 pregnant women in Belgium, Iceland, Denmark, Estonia, Norway, and Sweden demonstrated that 11% of pregnant women reported severe childbirth fear [29]. There were significant differences between the countries for prevalence of severe fear of childbirth, varying from 4.5% in Belgium to 15.6% in Estonia for primiparous women and from 7.6% in Iceland to 15.2% in Sweden for multiparous women.

Cultural norms regarding motherhood and birth shape women’s perceptions regarding what birth is and how it should be managed. Several studies have found differences between cultures in terms of women’s preferences regarding cesarean sections [30,31] and levels of fear of childbirth [29,32]. A systematic review of 490 full-text articles on the fear of childbirth was assessed for analysis [32]. It was found that various definitions and measurements of fear of childbirth were used. The most frequently used scale was the W-Delivery Expectancy/Experience Questionnaire with various cut-off points describing moderate, severe/intense, and extreme/phobic fear. Furthermore, rates of severe fear of childbirth measured in the same way varied in different countries. The reasons why the fear of childbirth might differ are unknown. 

In the current research, women who had never been pregnant, most often in the groups from Belarus, Poland, Greece, and Russia, reported fear more frequently, and in Turkey, excitement. Pregnant women from Belarus and Turkey reported excitement and fear more frequently than women from Poland and Greece. Women from Belarus and Turkey reported excitement in the postpartum group, as did women from Poland and Greece. When analyzing the responses of all the women from the five countries surveyed, fear was dominant at 46%.

In the present study, women frequently indicated excitement and joy evoked by pregnancy and childbirth. Our results are in accordance with previous studies [33,34,35]. When parents experience more significant meaning in life, satisfaction of their basic needs, more significant positive emotions, and enhanced social roles, they are happy and joyful [35]. Childbirth is an intense and transformative psychological experience that generates a sense of empowerment. The benefits of this process can be maximized through physical, emotional, and social support for women, enhancing their belief in their ability to give birth and not disturbing their physiology unless necessary.

To reduce the anxiety and stress associated with pregnancy and childbirth, women can attend childbirth classes to prepare them for motherhood and to help improve their psycho-physical condition. Contact with other pregnant women allows them to bond with women struggling with the same pregnancy symptoms or doubts.

Childbirth education classes are antenatal support services offered to pregnant women or couples to increase their knowledge regarding pregnancy, labor, delivery, breastfeeding, parenthood, and newborn care [12].

In the current study, most women surveyed did not attend childbirth classes, which may be related to the fact that schools are virtually non-existent or exclusively paid for in some countries.

Hildingsson et al. [36] researched the length of births without medical interventions between 2008 and 2013 in four Scandinavian countries. This study found that labor from the first uterine contractions to the birth of the child took an average of 14 h in primiparous women and 7.25 h in multiparous women. In the current study, labor lasted between ½ h and 42 h, including an average of 5.8 h in Belarus, an average of 5.6 h in Poland, an average of 10.8 h in Greece, an average of 4.6 h in Turkey, and an average of 5.3 h in Russia.

For several years, there has been a steady increase in cesarean sections [7]. The report by Boerma et al. [6] shows that the number of births by cesarean section increased from 12% in 2000 to 21% in 2015. According to data from the EURO-PERISTAT report [37], the rate of cesarean sections in 2014 in Europe was about 27%. In Poland, it accounted for 42.2% of births and was one of the highest rates in Europe. In the current study, in the case of pregnant women in Belarus and Russia, delivery was more often natural, while in Poland and Greece, delivery was more often by cesarean section. Overall, 630 women gave birth naturally and 283 by cesarean section.

The number of abortions worldwide is increasing. Between 2010 and 2014, it was around 56 million, an increase of 6 million compared to 1990–1994, meaning that almost one in four pregnancies ended in abortion [38]. However, the abortion rate, i.e., the number of abortions per 1000 women of reproductive age (15–44 years), decreased from 40 to 35. The abortion rate was found to be higher among married women, at 36 per 1000 women, against 25 per 1000 for unmarried women.

The current study found that no woman from Poland and Turkey had an induced abortion. In the pregnant group, 17 women in Belarus, 10 in Greece, and 3 in Russia had induced abortions, and in the postpartum group, 12 women in Belarus, 11 in Greece, and 2 in Russia had induced abortions.

An estimated 23 million miscarriages occur worldwide each year, which translates into 44 pregnancy losses per minute [39]. The overall risk of miscarriage is 15.3% of all identified pregnancies. The consequences of miscarriage are both physical, such as bleeding or infection, and psychological. Psychological consequences include increased risk of anxiety, depression, post-traumatic stress disorder, and suicide. In Poland, approximately 10–15% of all pregnancies end in miscarriage [40]. In the current study, in both the pregnant group and the postpartum group, women had between 1 and 3 natural miscarriages.

In the analysis of the relationships between the respondents’ associations regarding pregnancy and childbirth, the method of previous childbirth, and previous experience of natural and/or artificial miscarriage, it was found that the respondents very often had a feeling of fear during pregnancy and after childbirth.

The professional literature emphasizes that childbirth is the main factor causing anxiety for pregnant women, and it may have various origins. Birth anxiety may be caused mainly by social learning; seeing or hearing about the problems of other women during childbirth, women begin to fear the occurrence of similar events in their case [41]. Fear of childbirth significantly increases muscle tension and affects the perception of mental and somatic experiences, thus increasing pain, which is why reducing the fear of childbirth is so important [42]. Negative memories of previous births is another factor causing fear of the next one [43]. Anxiety may affect the course of childbirth, making labor difficult, contributing to obstetric complications, and increasing pain.

It should be emphasized that the experience of a traumatic childbirth may cause an acute stress reaction or post-traumatic stress disorder [44]. The current study showed a relationship between a previous cesarean section delivery and the association of anxiety with pregnancy and childbirth in the group of pregnant and postpartum women in Poland, and in the group of pregnant women with the experience of a previous cesarean or induced delivery in Turkey.

The study by Størksen et al. [43] confirms the thesis that women who feel more anxious are more likely to try to deliver by cesarean section, explaining this by previous traumatic experiences during childbirth. In turn, research by Demšar et al. [28] shows that women with a higher level of fear of childbirth prefer delivery by cesarean section. In comparison, women with a lower level of fear prefer vaginal delivery or the route of delivery is not important for them.

Reducing fear of childbirth involves a holistic approach that considers cultural, emotional, and medical aspects. By fostering trust and understanding, professionals can positively impact birthing experiences. There are many factors that can reduce the number of caesarean sections [45]. It is important to understand diverse cultural beliefs, practices, and expectations related to childbirth by medical staff. Health professionals should educate women on the benefits of natural delivery and the risks of cesarean sections. It is important to offer counseling and emotional support for pregnant women during prenatal visits. It is suggested that extra support for women should include sensitive education about the birth process, development of problem-solving skills, teaching coping strategies for labor, and affirming that negative childbirth events can be managed.

### Limitations of the Study

The study groups of women are relatively small for producing generalizable results. The women surveyed were from five different countries: Poland, Belarus, Russia, Greece, and Turkey. It is well known that cultural norms, values, and beliefs influence individuals’ behavior, decision-making, and perceptions. These factors can affect how people respond to surveys, questionnaires, or experimental conditions.It is worth assessing the felt anxiety using a standardized tool, such as the Childbirth Anxiety Questionnaire. Unfortunately, this tool was not validated in all surveyed countries, and the results could not be standardized.The study also did not include the form of childbirth preferred by the patients, which, as it turned out in the discussion, may not have depended on the fear of childbirth they felt, but instead on the expected form of childbirth.The sample size calculation was not done.The Cronbach’s α of the FAS scale is 0.643, which indicates that it is not high.

## 5. Implications 

Pregnancy and childbirth are two profound experiences that evoke a spectrum of emotions. Cultural norms regarding motherhood and birth shape women’s perceptions regarding what birth is and how it should be managed. Women’s experiences of pregnancy and childbirth can include all kinds of feelings, from joy to fear. Women’s fear of labor and birth is an essential reason for the increasing number of requests for and rates of cesarean sections. We suggest that doctors and midwives should educate expectant mothers that cesarean sections can be crucial in specific medical situations. However, unnecessary surgical procedures should be avoided to minimize risks for both mothers and babies.

It is also worth emphasizing that reducing anxiety during pregnancy may have a positive impact on the course of childbirth. Anxiety occurs in the population of pregnant women regardless of the expected form of delivery, and its level may affect their preferences as to the form of delivery. Prevention of fear of childbirth is very important and can be implemented through antenatal classes, as well as through social support received by the pregnant woman. We also cannot ignore medical issues or offer psychoeducation and organize a psychiatric consultation. This seems particularly important when deciding on the form of childbirth.

## 6. Conclusions

When thinking about pregnancy, respondents in all groups most often felt joy and excitement about childbirth as well as fear and excitement.

Natural miscarriage was experienced most frequently by women in Poland and Greece and induced in Belarus.

Generally, the women surveyed did not participate in childbirth classes.

The largest number of women gave birth naturally or by cesarean section in Poland and Belarus and the smallest number in Russia.

There were differences in all assessed aspects related to pregnancy and childbirth in the group of pregnant and postpartum women from Poland, Greece, Turkey, Belarus, and Russia between the study groups in each country.

With the exception of Russia, significant correlations were shown between the feeling of fear of pregnancy and childbirth, the way the previous/current pregnancy was delivered, and the experience of an artificial/natural miscarriage in the past.

## Figures and Tables

**Table 1 jcm-13-03681-t001:** Demographics of respondents.

	Belarus N = 386			Poland N = 467			Greece N = 364			Turkey N = 360			Russia N = 440		
	1	2	3	1	2	3	1	2	3	1	2	3	1	2	3
	N = 136	N = 147	N = 103	N = 131	N = 173	N = 163	N = 144	N = 114	N = 106	N = 150	N = 103	N = 107	N = 345	N = 47	N = 48
MARITAL STATUS															
Unmarried	117	5	10	26	14	42	89	3	2	88	0	0	319	0	3
Married	4	118	85	80	142	98	19	97	100	59	103	107	4	46	41
Divorced	3	4	4	5	2	5	0	9	2	2	0	0	0	1	1
Separated	1	0	0	0	2	3	1	1	0	0	0	0	2	0	1
Informal relationship	11	19	4	14	11	6	35	1	0	0	0	0	20	0	1
Widow	0	1	0	6	2	9	0	3	2	1	0	0	0	0	1
AGE (17–50 years)															
Average age	19.2 ± 19	29.2 ± 29	28.7 ± 29	33.5 ± 33	30.4 ± 30	31 ± 30	21 ± 20	34.7 ± 33	37.6 ± 36	28.3 ± 26	27.7 ± 27	28.1 ± 28	18.5 ± 18	27.7 ± 29	28.1 ± 28
PLACE OF RESIDENCE															
Rural	10	8	4	12	60	54	29	30	18	15	22	23	30	1	2
Urban	126	139	99	119	113	109	115	84	88	135	81	84	315	46	46
EDUCATION															
Vocational	4	41	37	7	11	11	1	18	15	51	36	42	5	16	22
Bachelor’s degree	2	35	16	33	28	23	27	33	32	74	40	41	4	12	8
Master’s degree	9	40	23	42	75	84	3	5	8	18	8	11	9	6	13
Student	92	12	10	14	10	13	109	3	4	1	1	13	250	10	2
Secondary	29	19	17	35	49	32	4	55	47	6	18	0	77	3	3
MATERIAL STATUS															
Very low	2	9	2	0	0	0	0	1	0	0	0	0	1	1	0
Low	5	4	7	1	2	4	1	9	7	3	20	23	11	3	3
Average	33	29	44	17	17	23	37	31	29	37	38	40	65	15	15
High	96	105	50	113	154	136	106	73	70	130	45	44	268	28	30
Average points	6.5 ± 2.0	6.3 ± 1.7	5.6 ± 1.7	7.9 ± 1.9	8.6 ± 2.5	7.6 ± 1.0	7.1 ± 2.0	6.6 ± 2.2	7.1 ± 1.2	5.8 ± 3.1	5.5 ± 2.3	5.2 ± 2.0	6.8 ± 1.7	5.6 ± 1.4	6.0 ± 1.5

1—women who have never given birth and are not pregnant, 2—pregnant women, and 3—postpartum women; Spearman’s rank correlation test.

**Table 2 jcm-13-03681-t002:** Associations of respondents regarding pregnancy.

Associations Regarding Pregnancy	Belarus N = 386			Poland N = 467			Greece N = 364			Turkey N = 360			Russia N = 440		
	1	2	3	1	2	3	1	2	3	1	2	3	1	2	3
	N = 136	N = 147	N = 103	N = 131	N = 173	N = 163	N = 144	N = 114	N = 106	N = 150	N = 103	N = 107	N = 345	N = 47	N = 48
Joy	42	130	94	108	145	140	96	102	106	35	56	102	70	44	45
Excitement	43 ^###^	86	62	70	91	88	60	72	77	38	75	89	80 ^###^	34	26
Fatigue	22	15	13	50	50	54	47	37	32	34	36	32	29	4	3
Helplessness	5	0	1	7	11	8	12	22	12	0	2	0	10	0	0
Fear	26	8	11	38	49	30	64	38	43	0	4	0	39	2	5
Hospital	3	3	0	26	37	19	20	88	8	0	2	5	20	0	1
House	0	0	0	13	29	14	16	3	9	24	16	10	4	0	0
Loneliness	2	0	181	1	0	0	2	3	5	1	7	2	2	0	80
Summary															
	Belarus			Poland		Greece			Turkey		Russia			Total	
Joy	266			393		304			193		159			1315	
Excitement	191			249		209			202		140			991	
Fatigue	50			154		116			102		36			458	
Helplessness	6			26		46			2		10			90	
Fear	45			117		145			4		46			357	
Hospital	6			82		116			7		21			232	
House	0			56		28			50		4			138	
Loneliness	183			1		10			10		82			286	

1—women who have never given birth and are not pregnant, 2—pregnant women, and 3—postpartum women (several answers could be chosen); Spearman’s rank correlation test (r = −0.480; ^###^ *p* < 0.001; r = −0.247; ^###^ *p* < 0.001).

**Table 3 jcm-13-03681-t003:** Diseases reported by respondents.

Diseases *	Belarus N = 250			Poland N = 336			Greece N = 220			Turkey N = 210			Russia N = 95		
	1	2	3	1	2	3	1	2	3	1	2	3	1	2	3
	N = 136	N = 147	N = 103	N = 131	N = 171	N = 163	N = 144	N = 114	N = 106	N = 150	N = 103	N= 107	N = 345	N = 47	N = 48
Diabetes	0	2	2	6	32	26	2	4	3	1	4	1	1	2	0
Hypertension	0	1	8	8	21	18	0	8	4	1	7	0	0	18	3
Anaemia	0	23 ^#^	25 ^#^	4	26	34	0	16	23	0	6	0	0	5	7
Spotting	0	6	19	0	19	15	0	1	5	0	25	0	0	3	3
Bleeding	0	7	8	0	10	8	0	11	14	0	7	0	0	0	0
Skin diseases	1	1	2	1	3	2	0	2	6	0	2	0	0	12	0
Infections	10	45 ^##^	43 ^##^	16	22	27	0	2	2	0	9	0	1	1	19
Infectious diseases	0	1	10	0	5	4	1	0	1	0	0	0	0	1	3

* several answers could be chosen; 1—women who have never given birth and are not pregnant, 2—pregnant women, 3—postpartum women; Spearman’s rank correlation test (^#^ r = 0.160, *p* = 0.01; ^##^ r = 0.497, *p* = 0.005 positive correlation with fear).

**Table 4 jcm-13-03681-t004:** Associations of respondents regarding childbirth.

Associations Regarding Childbirth *	Belarus N = 386			Poland N = 467			Greece N = 364			Turkey N = 360			Russia N = 440		
	1	2	3	1	2	3	1	2	3	1	2	3	1	2	3
	N = 136	N = 147	N = 103	N = 131	N = 173	N = 163	N = 144	N = 114	N = 106	N = 150	N = 103	N = 107	N = 345	N = 47	N = 48
Joy	11	62	37	42	37	46	43	61	67	5	12	54	20	17	19
Excitement	34	102 *	63	31	56	86	31	38	52	37	78 *	67 *	53	36	33
Fatigue	13	13	18	74	60	12	61	26	28	22	16	59	28	6	4
Helplessness	6	3	3	16	24	78	22	16	11	5	10	7	13	1	3
Fear	38	51	37	78	107	94	84 ^###^	67	69	3	52	12	72 ^###^	12	14
Hospital	30	46	26	69	102	0	59	36	38	0	45	6	58	16	14
House	1	1	5	3	1	34	4	3	4	0	0	25	3	0	2
Loneliness	0	0	0	0	0	0	2	1	4	9	7	2	0	3	0
Summary															
	Belarus		Poland			Greece			Turkey			Russia		Total	
Joy	110		125			171			138			123		667	
Excitement	199 ^###^		173			121			234 ^###^			174		901	
Fatigue	44		146			115			125			66		496	
Helplessness	12		118			49			33			28		240	
Fear	126		279			220			136			167		928	
Hospital	102		171			133			89			126		621	
House	7		38			11			29			9		94	
Loneliness	0		0			7			22			7		36	

1—women who have never given birth and are not pregnant; 2—pregnant women; 3—postpartum women * several answers could be chosen. Spearman’s rank correlation test, ^###^ *p* < 0.001 .

**Table 5 jcm-13-03681-t005:** Number of children.

Number of Children	Belarus N = 386			Poland N = 467			Greece N = 364			Turkey N = 360			Russia N = 440		
	1	2	3	1	2	3	1	2	3	1	2	3	1	2	3
	N = 136	N = 147	N = 103	N = 131	N = 173	N = 163	N = 144	N = 114	N = 106	N = 150	N = 103	N = 107	N = 345	N = 47	N = 48
0	136	22	7	131	38	15	144	1	1	150	28	22	345	18	3
1	0	66	41	0	62	63	0	32	19	0	31	34	0	17	18
2	0	50	37	0	47	62	0	53	64	0	19	17	0	7	19
3	0	9	16	0	17	18	0	20	15	0	12	16	0	2	5
4	0	0	1	0	6	4	0	6	4	0	7	9	0	2	2
5	0	0	1	0	1	1	0	2	3	0	6	9	0	1	1
The average number of children	0	1.4 ± 0.7	1.7 ± 0.9	0	1.4 ± 1.1	1.6 ± 0.9	0	2.0 ± 0.9	2.1 ± 0.9	0	3.1 ± 1.4	1.8 ± 1.5	0	1.1 ± 1.2	1.8 ± 1.0
Summary															
	Belarus			Poland			Greece		Turkey			Russia		Total	
0	165			185			146		200			366		1062	
1	107			125			51		65			35		383	
2	87			109			117		36			26		375	
3	25			35			35		28			7		130	
4	1			10			10		16			4		41	
5	1			3			5		15			2		26	

1—women who have never given birth and are not pregnant; 2—pregnant women; 3—postpartum women. Independent samples *t*-test *p* < 0.001 vs. other groups (Belarus, Poland, Greece, and Russia).

**Table 6 jcm-13-03681-t006:** Number of births to date survey.

Number of Births to Date	Belarus N = 250		Poland N = 308		Greece N = 230		Turkey N = 210		Russia N = 95	
	1	2	1	2	1	2	1	2	1	2
	N = 147	N = 103	N = 173	N = 163	N = 114	N = 106	N = 103	N = 107	N = 47	N = 48
0	81	18	78	58	27	4	60	10	22	21
1	37	22	42	17	15	10	25	62	13	3
2	24	36	34	59	54	70	11	25	12	17
3	4	16	12	21	13	13	7	7	0	3
4	0	1	4	8	4	8	0	3	0	4
5	1	1	1	0	1	1	0	0	0	0
			Summary							
	Belarus		Poland	Greece		Turkey	Russia		Total	
0	105		136	41		70	43		389	
1	65		59	25		87	16		252	
2	63		75	124		36	29		327	
3	20		25	26		14	3		88	
4	1		12	12		3	4		32	
5	2		1	2		0	0		5	
sum	250		308 ^###^	230		210	95			

1—pregnant women; 2—postpartum women, Chi-square test, ^###^ *p* < 0.001 vs. Russia.

**Table 7 jcm-13-03681-t007:** Type of last delivery.

Type of Delivery	Belarus N = 250		Poland N = 308		Greece N = 230		Turkey N = 210		Russia N = 95	
	1	2	1	2	1	2	1	2	1	2
	N = 147	N = 103	N = 173	N = 163	N = 114	N = 106	N = 103	N = 107	N = 47	N = 48
Natural	29	32	28	35	27	34	16	34	11	18
Induced	7	16	25	29	23	25	8	21	4	6
Caesarean section	27	34	39	40	33	39	19	42	7	3
Vacuum	3	3	1	1	4	4	0	0	3	1
Total	66	85	93	105	87	102	43	97	25	27
			Summary							
	Belarus		Poland	Greece		Turkey		Russia		Total
Natural	61		63	61		50		29		264
Induced	23		54	48		29		10		164
Caesarean section	61		79	72 ^##^		61		10		283
Vacuum	6		2	8		0		4		20

1—pregnant women; 2—postpartum women. Chi-square test, ^##^ *p* = 0.002 vs. Russia.

**Table 8 jcm-13-03681-t008:** History of spontaneous and induced abortions.

	Belarus N = 250		Poland N = 308		Greece N = 230		Turkey N = 210		Russia N = 95	
	1	2	1	2	1	2	1	2	1	2
	N = 147	N = 103	N = 173	N = 163	N = 114	N = 106	N = 103	N = 107	N = 47	N = 48
			Spontaneous abortions							
None	125	87	134	129	87	75	74	107	38	40
Yes	22	16	37	34	27	31	29	0	9	8
One miscarriage	19	14	30	26	25	26	17	0	9	8
Two miscarriages	1	2	4	4	2	5	11	0	0	0
Three miscarriages	2	0	3	4	0	0	1	0	0	0
Total	147	103	171	163	114	106	103	107	47	48
			Induced abortions							
	Belarus N = 250		Poland N = 308		Greece N = 230		Turkey N = 210		Russia N = 95	
	1	2	1	2	1	2	1	2	1	2
	N = 147	N = 103	N = 173	N = 163	N = 114	N = 106	N = 103	N = 107	N = 47	N = 48
None	130	91	167	158	104	95	94	107	44	46
Yes	17	12	6	5	10	11	9	0	3	2
One miscarriage	12	10	4	5	7	7	9	0	2	2
Two miscarriages	3	1	2	0	3	2	0	0	1	0
Three miscarriages	2	0	0	0	0	2	0	0	0	0
Five miscarriages	0	1	0	0	0	0	0	0	0	0
Total	147	103	171	163	114	106	103	107	47	48

1—pregnant women; 2—postpartum women.

**Table 9 jcm-13-03681-t009:** Respondents’ associations regarding pregnancy and the method of previous/current delivery.

Associations Regarding Pregnancy	Type of Delivery/ *p*-Value and R-Value	Belarus N = 250		Poland N = 308		Greece N = 230		Turkey N = 210		Russia N = 95	
		1	2	1	2	1	2	1	2	1	2
joy	Natural	NS	NS	NS	NS	NS	NS	NS	NS	NS	NS
	Induced	NS	NS	NS	NS	NS	NS	NS	NS	NS	NS
	Caesarean section	NS	NS	NS	NS	NS	NS	NS	NS	NS	NS
	Vacuum	NS	NS	NS	NS	NS	NS	There was no	There was no	NS	NS
excitement	Natural	NS	NS	R = 0.254 *p* = 0.008	NS	NS	NS	NS	NS	NS	NS
	Induced	NS	NS	NS	NS	NS	NS	NS	NS	NS	NS
	Caesarean section	NS	NS	NS	NS	NS	NS	NS	NS	NS	NS
	Vacuum	NS	NS	NS	NS	NS	NS	There was no	There was no	NS	NS
tiredness	Natural	NS		NS			NS	NS	NS	NS	NS
	Induced	NS	NS	NS	NS	NS	NS	NS	NS	NS	NS
	Caesarean section	NS	NS	NS	NS	NS	NS	NS	NS	NS	NS
	Vacuum	NS	NS	NS	NS	NS	NS	There was no	There was no	NS	NS
helplessness	Natural	NS		NS			NS	NS	NS	NS	NS
	Induced	NS	NS	NS	NS	NS	NS	NS	NS	NS	NS
	Caesarean section	NS	NS	NS	R = −0.193 *p* = 0.046	NS	NS	NS	NS	NS	NS
	Vacuum	NS	NS	NS		NS	NS	There was no	There was no	NS	NS
fear	Natural	R = −0.441 *p*= 0.0002	R = −0.451 *p* = 0.0001	R = −0.207 *p* = 0.032	NS	NS	R = −0.377 *p* = 0.000	NS	NS	NS	NS
	Induced	NS	NS	NS	NS	NS	NS	R = −0.706 *p* = 0.000	NS	NS	NS
	Caesarean section	NS	NS	R = −0.266 *p* = 0.006	NS	NS	NS	R = −0.312 *p* = 0.039	NS	NS	NS
	Vacuum	NS	NS	NS	NS	NS	NS	There was no	There was no	NS	NS
loneliness	Natural	NS	NS	NS	NS		NS	NS	NS	NS	NS
	Induced	NS	NS	R = −0.251 *p* = 0.026	NS	NS	NS	NS	NS	NS	NS
	Caesarean section	NS	NS	NS	NS	NS	NS	NS	NS	NS	NS
	Vacuum	NS	NS	NS	NS	NS	NS	There was no	There was no	NS	NS

1—pregnant women; 2—postpartum women. Spearman’s rank correlation test R, *p* values, NS—not significant.

**Table 10 jcm-13-03681-t010:** Respondents’ associations regarding pregnancy between women who have and who have not had a natural miscarriage in the past.

Type of Miscarriage/*p*-Value and R-Value	Belarus N = 250		Poland N = 308		Greece N = 230		Turkey N = 210		Russia N = 95	
	1	2	1	2	1	2	1	2	1	2
joy	NS	NS	NS	NS	NS	NS	There were no natural miscarriages		NS	NS
excitement	NS	NS	NS	NS	NS	NS			R = 0.388 *p* = 0.007	NS
tiredness	NS	NS	NS	NS	NS	NS			NS	NS
helplessness	R = −0.571 *p* < 0.001	R = −0.492 *p* < 0.001	NS	NS	NS	NS			NS	NS
fear	R = −0.572 *p* < 0.001	R = −0.266 *p* = 0.001	R = −0.246 *p* = 0.001	NS	R = −0.223 *p* = 0.017	NS			NS	NS
loneliness	R = −0.396 *p* < 0.001	NS	R = −0.310 *p* < 0.001	NS	NS	NS			NS	NS

1—pregnant women; 2—postpartum women. Spearman’s rank correlation test R, *p* values, NS—not significant.

**Table 11 jcm-13-03681-t011:** Respondents’ associations regarding childbirth between women who have and who have not had an artificial miscarriage in the past.

Type of Miscarriage/*p*- Value and R-Value	Belarus N = 250		Poland N = 308		Greece N = 230		Turkey N = 210		Russia N= 95	
	1	2	1	2	1	2	1	2	1	2
joy	R = 0.228 *p* = 0.02	NS	NS	NS	NS	NS	There were no artificial miscarriages		NS	NS
excitement	NS	NS	NS	NS	NS	R = 0.205 *p* = 0.035			NS	NS
tiredness	R = −0.341 *p* < 0.001	NS	NS	NS	NS	R = −0.286 *p* = 0.003			NS	NS
helplessness	NS	NS	NS	NS	NS	NS			NS	NS
fear	NS	NS	NS	NS	NS	NS			R = −0.551 *p* < 0.00	NS
loneliness	NS	NS	NS	NS	NS	NS			NS	NS

1—pregnant women; 2—postpartum women. Spearman’s rank correlation test R, *p* values, NS—not significant.

**Table 12 jcm-13-03681-t012:** Associations of respondents regarding childbirth and the method of earlier childbirth.

Associations Regarding Pregnancy *	Type of Delivery/ *p*-Value and R-Value	Belarus N = 250		Poland N = 308		Greece N = 230		Turkey N = 210		Russia N = 95	
		1	2	1	2	1	2	1	2	1	2
joy	Natural	NS	NS	NS	NS	NS	NS	NS	NS	NS	NS
	Induced	NS	NS	NS	NS	NS	NS	NS	NS	NS	NS
	Caesarean section	NS	NS	NS	NS	NS	NS	NS	NS	NS	NS
	Vacuum	NS	NS	NS	NS	NS	NS	NS	NS	NS	NS
excitement	Natural	NS	NS	R = 0.254 *p* = 0.008	NS	NS	NS	NS	NS	NS	NS
	Induced	NS	NS	NS	NS	NS	NS	NS	NS	NS	NS
	Caesarean section	NS	NS	NS	NS	NS	NS	NS	NS	NS	NS
	Vacuum	NS	NS	NS	NS	NS	NS	NS	NS	NS	NS
tiredness	Natural	NS	NS	NS	NS	NS	NS	NS	NS	NS	NS
	Induced	NS	NS	NS	NS	NS	NS	NS	NS	NS	NS
	Caesarean section	NS	NS	NS	NS	NS	NS	NS	NS	NS	NS
	Vacuum	NS	NS	NS	NS	NS	NS	NS	NS	NS	NS
helplessness	Natural	NS	NS	NS	NS	NS	NS	NS	NS	NS	NS
	Induced	NS	NS	NS	NS	NS	NS	NS	NS	NS	NS
	Caesarean section	NS	NS	NS	R = 0.193 *p* = 0.046	NS	NS	NS	NS	NS	NS
	Vacuum	NS	NS	NS	NS	NS	NS	NS	NS	NS	NS
fear	Natural	NS	NS	R = −0.207 *p* = 0.032	NS	NS	NS	NS	NS	NS	NS
	Induced	NS	NS	NS	NS	NS	NS	NS	NS	NS	NS
	Caesarean section	NS	NS	R = 0.222 *p* = 0.021	NS	NS	NS	NS	NS	NS	NS
	Vacuum	NS	NS	NS	NS	NS	NS	NS	NS	NS	NS
loneliness	Natural	NS	NS	NS	NS	NS	NS	NS	NS	NS	NS
	Induced	NS	NS	NS	NS	NS	NS	NS	NS	NS	NS
	Caesarean section	NS	NS	NS	NS	NS	NS	NS	NS	NS	NS
	Vacuum	NS	NS	NS	NS	NS	NS	NS	NS	NS	NS

1—pregnant women; 2—postpartum women. Spearman’s rank correlation test R, *p* values. * several answers could be chosen.

## Data Availability

The data that support the findings of this study are not openly available due to reasons of sensitivity and are available from the corresponding author upon reasonable request.

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
