# Peer review of "Pregnancy and Childbirth Fear of Women from Poland, Greece, Turkey, Belarus, and Russia"

_jcm, 2024, doi:10.3390/jcm13133681_

Round 1

Reviewer 1 Report

Comments and Suggestions for Authors

1.      The term “perception” in the title may be vague, the term Joy / Fear / Emotion / Excitement may be included in the title ( It is a suggestion not mandatory )

2.      The inclusion & Exclusion criteria is well defined but alcoholic or smokers could have been excluded or group them as an exclusive group.

3.      Additionally , first time pregnant women data can be analysed exclusively

4.      Did Economic status or occupation or family support or education played any role?

5.      Over all the data were well tabulated and analysed statistically proven as well

Author Response

The term “perception” in the title may be vague, the term Joy / Fear / Emotion / Excitement may be included in the title ( It is a suggestion not mandatory )

We have changed Perception on Fear in the title.

  1. The inclusion & Exclusion criteria is well defined but alcoholic or smokers could have been excluded or group them as an exclusive group.

Yes, you are right. However, this study did not include these variables in the inclusion and exclusion criteria.

  1. Additionally , first time pregnant women data can be analysed exclusively.

Thank you for your suggestion. In this study, we did not analyse the first-time pregnant women's data. This is a good topic for further research

  1. Did Economic status or occupation or family support or education played any role?

Yes, you are right. We have analysed the material status.  We did not analyse  other variables  which may play the role.

  1. Over all the data were well tabulated and analysed statistically proven as well.

Thank you for your coment.

Reviewer 2 Report

Comments and Suggestions for Authors

Dear Authors:

Thank you for allowing me to review your manuscript. The topic is very important and conducting multicenter studies in several countries can help to improve maternal care for women in many countries. I am going to make multiple comments whose sole purpose is to try to improve the manuscript.

Introduction: Can be improved. The relevance of some references could be improved. In addition, in certain parts of the introduction, statements are made that are not adequately supported by references. I will give you just a few examples:

In recent years, Poland has seen a change in fertility patterns, a decline in the number of births in all reproductive age groups, and a decrease in the absolute number of births [5]. The fertility rate for Poland in 2023 was 1.467 births per woman, a 0.62% increase from 2022. The fertility rate for Belarus in 2023 was 1.736 births per woman, a 0.29% increase from 2022. The fertility rate for Russia in 2023 was 1.825 births per woman, a 0.05% increase from 2022. The birth rate for Greece in 2023 was 6.930 births per 1000 women, a 2.56% decline from 2022. The fertility rate for Turkey in 2023 was 1.994 births per woman, a 0.85% decline from 2022.

Also, fear of childbirth may affect the birth rate. Fear of childbirth is a natural reaction for many expectant mothers. Women often experience uncertainty about the labor process. They often fear that something might go wrong or be painful. Pregnant women are increasingly exposed to the view that pregnancy and childbirth are intrinsically dangerous, requiring medical monitoring and management

It is a specific anxiety disorder where individuals experience an irrational and excessive fear related to pregnancy and childbirth. Women with this phobia may avoid becoming pregnant altogether or opt for a cesarean section to avoid vaginal birth. Some women may experience Sleep disturbances, panic attacks, and nightmares

Concrete references should be provided in these cases. On the other hand, the sentence: “In 2021, the rate was 1.33, thus below the EU average and well below the replacement level [8]” is based on a reference to a Korean study that does not seem to correspond. The same occurs in “In a Greek study [20], Validation of the Childbirth Attitudes Questionnaire, 145 late-pregnancy women participated.” I think it should be citation 19. This occurs in many other parts of the manuscript.

I suggest the authors to check the citations, this type of error may have occurred after a previous revision of the manuscript.

Method

-Sample size: Indicate what sample calculation has been performed. A sample calculation should have been performed for a population estimate based on a proportion or an expected standard deviation.

-What does GP clinics mean?

-You refer to a control group. Can you explain this? The study design does not imply that there has to be a control group.

-Willing to give informed consent should not be considered as inclusion criteria. And not give informed consent should be considered withdrawal criteria, not exclusion criteria.

- You should indicate in the methods which variables were collected, since later in the results, data and results appear for variables not mentioned in the method.

-Regarding the measurements performed. From my point of view, this is the main problem with this work. You have attempted to analyze factors that may affect the perception of pregnancy and childbirth, with special emphasis on the fear of childbirth. But to evaluate this perception and to be able to relate it to factors a validated tool should be used and you seem to have used a questionnaire that only asks 8 variables (joy, excitement, fatigue, helplessness, fear, hospital, house, loneliness) based on yes or no answers (or so I understand). Considering, for example, that there are many validated instruments to measure fear of childbirth, expectations of childbirth, satisfaction with childbirth, etc., using an instrument that has not been validated and for which there is no reliability data is at least questionable. This is a very weak point of your study, which affects the validity of the results. For example, what does hospital or house mean in this instrument? The only reliability data on an instrument used in your research is on the Family Affluence Scale (FAS) (again citation 21 does not apply) and it reports a low cronbach's coefficient, below 0.70, which indicates low reliability (internal consistency). Extensive justification should be provided regarding the use of a non-validated tool and the use of this questionnaire should be explained in detail. I understand that the choice of measuring instrument in an international study is a handicap, but this should not prevent the use of an instrument that provides reliable data.

In the analysis you refer that you have used Student's t-test. Indicate which test you performed to evaluate the normality of the distribution and therefore justify the convenience of a parametric test. On the other hand, there are many groups in your study: did you not carry out any comparison of means in more than two groups? In the results there are results that I understand to be correlations, but you do not explain this in the analysis that appears in the method.

-I can't find any analysis that relates perceptions with diseases, when it is a key factor that can affect the fear of childbirth.

I think there is much margin for improvement in the tables and in the presentation of results. It should be possible to appreciate in these the inferences made, providing all the p-values, both statistically significant and non-significant, and being able to evaluate which statistical test has been used in each specific case. 

I encourage the authors to carry out a deep revision of the manuscript. Best regards

Author Response

Reviewer 2

Dear Authors:

Thank you for allowing me to review your manuscript. The topic is very important and conducting multicenter studies in several countries can help to improve maternal care for women in many countries. I am going to make multiple comments whose sole purpose is to try to improve the manuscript.

Introduction: Can be improved. The relevance of some references could be improved. In addition, in certain parts of the introduction, statements are made that are not adequately supported by references. I will give you just a few examples:

In recent years, Poland has seen a change in fertility patterns, a decline in the number of births in all reproductive age groups, and a decrease in the absolute number of births [5]. The fertility rate for Poland in 2023 was 1.467 births per woman, a 0.62% increase from 2022. The fertility rate for Belarus in 2023 was 1.736 births per woman, a 0.29% increase from 2022. The fertility rate for Russia in 2023 was 1.825 births per woman, a 0.05% increase from 2022. The birth rate for Greece in 2023 was 6.930 births per 1000 women, a 2.56% decline from 2022. The fertility rate for Turkey in 2023 was 1.994 births per woman, a 0.85% decline from 2022.

Also, fear of childbirth may affect the birth rate. Fear of childbirth is a natural reaction for many expectant mothers. Women often experience uncertainty about the labor process. They often fear that something might go wrong or be painful. Pregnant women are increasingly exposed to the view that pregnancy and childbirth are intrinsically dangerous, requiring medical monitoring and management

It is a specific anxiety disorder where individuals experience an irrational and excessive fear related to pregnancy and childbirth. Women with this phobia may avoid becoming pregnant altogether or opt for a cesarean section to avoid vaginal birth. Some women may experience Sleep disturbances, panic attacks, and nightmares

Concrete references should be provided in these cases. On the other hand, the sentence: “In 2021, the rate was 1.33, thus below the EU average and well below the replacement level [8]” is based on a reference to a Korean study that does not seem to correspond. The same occurs in “In a Greek study [20], Validation of the Childbirth Attitudes Questionnaire, 145 late-pregnancy women participated.” I think it should be citation 19. This occurs in many other parts of the manuscript.

Thank you for your remarks, we have corrected citation of the references.

I suggest the authors to check the citations, this type of error may have occurred after a previous revision of the manuscript.

Method

-Sample size: Indicate what sample calculation has been performed. A sample calculation should have been performed for a population estimate based on a proportion or an expected standard deviation.

We did not calculate the sample size. It was difficult time  (the COVID-19 pandemic) to perform studies.

-What does GP clinics mean? General Practice Clinics. We have corrected it in the text

-You refer to a control group. Can you explain this? The study design does not imply that there has to be a control group.

This is our mistake. We did not have the control group.  We have corrected it.    the women who had never given birth and were not pregnant, this group included female students and university workers in the studied countries

-Willing to give informed consent should not be considered as inclusion criteria. And not give informed consent should be considered withdrawal criteria, not exclusion criteria.

With suggestions, we have removed it from the text.

- You should indicate in the methods which variables were collected, since later in the results, data and results appear for variables not mentioned in the method.

In the Measures, we mentioned that in the original questionnaire analysed : age, marital status, place of residence, education, material status, and associations of respondents regarding pregnancy and childbirth (joy, excitement, fatigue, helplessness, fear, hospital, house, loneliness) are associated with, and health problems in the last year. In the group of pregnant and postpartum women, the respondents were additionally asked about the number of children, spontaneous and induced abortions, duration of the delivery, total number of deliveries, type of the last delivery, and participation in childbirth classes.

-Regarding the measurements performed. From my point of view, this is the main problem with this work. You have attempted to analyze factors that may affect the perception of pregnancy and childbirth, with special emphasis on the fear of childbirth. But to evaluate this perception and to be able to relate it to factors a validated tool should be used and you seem to have used a questionnaire that only asks 8 variables (joy, excitement, fatigue, helplessness, fear, hospital, house, loneliness) based on yes or no answers (or so I understand). Considering, for example, that there are many validated instruments to measure fear of childbirth, expectations of childbirth, satisfaction with childbirth, etc., using an instrument that has not been validated and for which there is no reliability data is at least questionable. This is a very weak point of your study, which affects the validity of the results. For example, what does hospital or house mean in this instrument? The only reliability data on an instrument used in your research is on the Family Affluence Scale (FAS) (again citation 21 does not apply) and it reports a low cronbach's coefficient, below 0.70, which indicates low reliability (internal consistency). Extensive justification should be provided regarding the use of a non-validated tool and the use of this questionnaire should be explained in detail. I understand that the choice of measuring instrument in an international study is a handicap, but this should not prevent the use of an instrument that provides reliable data.

Yes, you are right; we used  not-validated questionnaire. It is worth assessing the felt anxiety using a standardized tool, such as the Childbirth Anxiety Questionnaire, but unfortunately, this tool was not validated in all surveyed countries, and the results could not be standardized. This is a weak point of this study. We have mentioned it in the study's limitations.

In the analysis you refer that you have used Student's t-test. Indicate which test you performed to evaluate the normality of the distribution and therefore justify the convenience of a parametric test. On the other hand, there are many groups in your study: did you not carry out any comparison of means in more than two groups? In the results there are results that I understand to be correlations, but you do not explain this in the analysis that appears in the method.

The Shapiro-Wilk test was used for the test of normality. We used test -t was used to compare the means of the two groups. We have compared number of children.

-I can't find any analysis that relates perceptions with diseases, when it is a key factor that can affect the fear of childbirth.

We have performed this analysis. A positive correlation was found between anemia, infections, and fear in women from Belarus. Spearman's rank correlation test, (#r= 0.160, p=0,01; ## r= 0.497, p=0.005 positive correlation with fear. Table 3.

I think there is much margin for improvement in the tables and in the presentation of results. It should be possible to appreciate in these the inferences made, providing all the p-values, both statistically significant and non-significant, and being able to evaluate which statistical test has been used in each specific case. 

We have corrected it.

I encourage the authors to carry out a deep revision of the manuscript. Best regards

Reviewer 3 Report

Comments and Suggestions for Authors

Dear authors

The research study "Pregnancy and childbirth in the perception of women in Poland, Greece, Turkey, Belarus and Russia" focused on a relevant objective because understanding women's perceptions of care during pregnancy and childbirth continues to be a challenge for health professionals and policy makers, who seek to improve obstetric care based on women's needs.

I have some concerns with this study as although the sample is large and highly diverse, the design has some significant flaws, including the study justification and objectives. I send some suggestions to improve the final quality of the manuscript.

Abstract: 

The objective of the study must be clarified, as perceptions other than fear are studied and must coincide in the summary and introduction. Only descriptive data is presented in the results topic. In the conclusions, the authors present data from the correlational analysis, but do not address the objective of the study. 

Introduction:

The introduction tries to cover many aspects such as the importance of motherhood, physical and emotional challenges, fear of childbirth, cesarean section trends, and fertility rates. However, the logical sequence of information needs to be improved.      For example, moving from discussing the general meaning of motherhood to the specifics of C-section trends and fertility rates needs to be more cohesive. 

    The research studies other perceptions about pregnancy and childbirth; however, this approach is not adopted in the introduction.

Authors must also improve their arguments about the importance of the study.

    Consider moving some of the statistical data presented in the last paragraph of the introduction to the discussion section, where it makes more sense.

Methodology:

    It would be helpful to include subchapters to guide readers through the methodology.

When selecting the sample, we suggest that you justify the choice of countries that are part of the study, as well as the sample size of each country.     In addition to including details on data collection, distribution and collection of questionnaires, considering that the study took place during the COVID-19 pandemic.

The authors should also clarify why they use a comparative group (n= 902) of students who have never been pregnant.

Statistical analysis was performed using correlations. However, this aspect is not mentioned in the statistical analysis.

Results:

    The tables are numerous (there must be a maximum of six tables) and must include percentages and inferential analysis data.

The title of Table 2, “Respondents' associations in relation to pregnancy”, does not seem appropriate since the authors only present descriptive data. The same situation is repeated in Table 4. 

Page 8 - "In total, significantly (p<0.001) more often, 630 women gave birth naturally (???) than 283 women by cesarean section. Details are shown in Table 7." The data in the table does not agree with the information in this paragraph.

Discussion

The discussion is supported by some studies dating back more than 20 years and from the North American context. Given the significant changes in maternal and obstetric health care over the past two decades and the cultural realities of the countries included in the study, it seems inappropriate. We suggest including more current studies that are closer to the geographic context under study.

In the discussion, it is essential to delve deeper into the implications of the results, particularly the sociocultural differences between the five countries studied, as they certainly present very different realities in prenatal care and childbirth.

Including more practical recommendations based on the results can significantly improve the discussion. For example, how can maternal and obstetric health professionals use this information to reduce fear of childbirth and cesarean section rates?

The authors could also more explicitly address possible biases due to the COVID-19 pandemic, as it would be important to understand whether fear and anxiety were related to the obstetric situation or the pandemic reality. 

Many of the results were not discussed consistently.

References:

Include some more recent studies, mainly to support the discussion.

Author Response

Reviewer 3

Dear authors

The research study "Pregnancy and childbirth in the perception of women in Poland, Greece, Turkey, Belarus and Russia" focused on a relevant objective because understanding women's perceptions of care during pregnancy and childbirth continues to be a challenge for health professionals and policy makers, who seek to improve obstetric care based on women's needs.

I have some concerns with this study as although the sample is large and highly diverse, the design has some significant flaws, including the study justification and objectives. I send some suggestions to improve the final quality of the manuscript.

Abstract: 

The objective of the study must be clarified, as perceptions other than fear are studied and must coincide in the summary and introduction. Only descriptive data is presented in the results topic. In the conclusions, the authors present data from the correlational analysis, but do not address the objective of the study. 

With suggestions we have corrected the Abstract section.

Introduction:

The introduction tries to cover many aspects such as the importance of motherhood, physical and emotional challenges, fear of childbirth, cesarean section trends, and fertility rates. However, the logical sequence of information needs to be improved.      For example, moving from discussing the general meaning of motherhood to the specifics of C-section trends and fertility rates needs to be more cohesive. 

    The research studies other perceptions about pregnancy and childbirth; however, this approach is not adopted in the introduction.

Authors must also improve their arguments about the importance of the study.

We have provided more data.

Numerous studies have been published on fear of childbirth has a significant impact on delivery outcomes resulting in a dramatic increase in cesarean births [6,9,15,16, 17,18,19] 

It is suggested that some women are more susceptible to fear of childbirth than others. The etiology of tokophobia is multifactorial and can be associated with different factors such as susceptibility to anxiety or depression [16] and other psychiatric disorders, well-being in interpersonal relationships [17], and traumatic experiences of former birth [18]. Some women described themselves as lonely or with low self-esteem [19]. The need for psychiatric care and the presence of traumatic stress symptoms are reported outcomes together with prolonged labour, longer labours, use of epidural, and obstetric complications.

    Consider moving some of the statistical data presented in the last paragraph of the introduction to the discussion section, where it makes more sense.

In my opinion in other reviewers statistical data are more appropriate in the Introduction section

Methodology:

    It would be helpful to include subchapters to guide readers through the methodology.

When selecting the sample, we suggest that you justify the choice of countries that are part of the study, as well as the sample size of each country.     In addition to including details on data collection, distribution, and collection of questionnaires, considering that the study took place during the COVID-19 pandemic.

The authors should also clarify why they use a comparative group (n= 902) of students who have never been pregnant.

Clearly, this is not a comparative group; these are women who have never been pregnant. We have removed the word comparative group.

Statistical analysis was performed using correlations. However, this aspect is not mentioned in the statistical analysis.

We have provided it in the statistical analysis. The correlations between variables were calculated using Spearman’s rank correlation analysis. We used the interpretations of the r values: poor correlation is r=0.1-0.2; fair r=0.3-0.5; moderate r=0.6-0.7; very strong r=0.8-0.9.

Results:

    The tables are numerous (there must be a maximum of six tables) and must include percentages and inferential analysis data.

We have a lot of data from 5 countries, and it is difficult to reduce the number of tables.

If the editor agrees, 6 tables may be moved into the supplementary file.  

The title of Table 2, “Respondents' associations in relation to pregnancy”, does not seem appropriate since the authors only present descriptive data. The same situation is repeated in Table 4. 

In the description Table 2 there is   .Among the women who have never been pregnant, those from Belarus and Russia most often (p<0.001) reported excitement (n=43, n=80, respectively), (Table 2)

Similarly, we have the statistics in Table 4.

Page 8 - "In total, significantly (p<0.001) more often, 630 women gave birth naturally (???) than 283 women by cesarean section. Details are shown in Table 7." The data in the table does not agree with the information in this paragraph.

We have corrected it. In total, a little more than 283 women gave birth by cesarean section than 264 women by naturally.

Discussion

The discussion is supported by some studies dating back more than 20 years and from the North American context. Given the significant changes in maternal and obstetric health care over the past two decades and the cultural realities of the countries included in the study, it seems inappropriate. We suggest including more current studies that are closer to the geographic context under study.

We have included references from Europe, and we have removed reference from North America.

A European study of 6870 pregnant women in Belgium, Iceland, Denmark, Estonia, Norway, and Sweden demonstrated that 11% of pregnant women reported severe childbirth fear [27]. There were significant differences between the countries for prevalence of severe fear of childbirth, varying from 4.5% in Belgium to 15.6% in Estonia for primiparous women and from 7.6% in Iceland to 15.2% in Sweden for multiparous women.

Cultural norms regarding motherhood and birth shape women’s perceptions regarding what birth is and how it should be managed. Several studies have found differences between cultures in terms of women’s preferences regarding cesarean section [28, 29], and levels of fear of childbirth  [27,30]. A systematic review of 490 full-text articles on the fear of childbirth was assessed for analysis [30]. It was found that various definitions and measurements of fear of childbirth were used. The most frequently used scale was the W-Delivery Expectancy/Experience Questionnaire with various cut-off points describing moderate, severe/intense, and extreme/phobic fear. Furthermore, rates of severe fear of childbirth measured in the same way varied in different countries. The reasons why fear of childbirth might differ are unknown. 

In the discussion, it is essential to delve deeper into the implications of the results, particularly the sociocultural differences between the five countries studied, as they certainly present very different realities in prenatal care and childbirth.

The subject of sociocultural differences in prenatal care and childbirth is both interesting and important. The World Health Organization (WHO) recommends “culturally appropriate” maternity care services to improve maternal and newborn health.

Jones E., Lattof S.R. , Coast E. Interventions to provide culturally-appropriate maternity care services: factors affecting implementation. BMC Pregnancy Childbirth  2017,17, 267, 1-10. doi.org/10.1186/s12884-017-1449-7

Women use skilled maternity services more when they receive care that suits their culture. Implementation factors include economic and social factors, community participation, and person-centered care. It is well known, that timely, equitable access to quality prenatal care contributes to better health outcomes for birthing individuals and infants.

Including more practical recommendations based on the results can significantly improve the discussion. For example, how can maternal and obstetric health professionals use this information to reduce fear of childbirth and cesarean section rates?

We have provided more details on how reduce fear of childbirth and cesarean section rates.

Reducing fear of childbirth involves a holistic approach that considers cultural, emotional, and medical aspects. By fostering trust and understanding, professionals can positively impact birthing experiences. There are many factors that can reduce the number of caesarean sections. It is important to understand diverse cultural beliefs, practices, and expectations related to childbirth by medical staff. Health professionals should educate women on the benefits of natural delivery and the risks of cesarean sections. It is important to offer counseling and emotional support for pregnant women during prenatal visits. It is suggested that extra support for women should include sensitive education about the birth process, development of problem‐solving skills, teaching coping strategies for labor, and affirming that negative childbirth events can be managed.

O'Connell MA, Khashan AS, Leahy-Warren P, Stewart F, O'Neill SM. Interventions for fear of childbirth including tocophobia. Cochrane Database Syst Rev. 2021, 7(7):CD013321. doi: 10.1002/14651858.CD013321.pub2. 

 The authors could also more explicitly address possible biases due to the COVID-19 pandemic, as it would be important to understand whether fear and anxiety were related to the obstetric situation or the pandemic reality. 

In the Discussion section, we have written about the COVID-19 pandemic's impact on the level of fear and its effects on our results.

Our study was performed during the COVID-19 pandemic, a difficult time for all people, particularly pregnant women. The COVID-19 epidemic has led to increased fear, stress, and anxiety among pregnant women. They often perceive childbirth during this time as a threat to their well-being and health [22,23,24]. A study funded by the National Institutes of Health found that women who gave birth during the pandemic, especially in communities with COVID-19 outbreaks, were more likely to experience traumatic childbirth. This included symptoms of intense anxiety or post-traumatic stress disorder triggered by the birthing experience [25]. Pregnant women were very often worried about the virus’s impact on their pregnancy and unborn babies. These negative factors could also impact on our results.

Many of the results were not discussed consistently.

References:

Include some more recent studies, mainly to support the discussion.

We have provided more recent references.

Reviewer 4 Report

Comments and Suggestions for Authors

Thank you for the efforts you put in your research. Working with big data in such condition (COVID-19 pandemic) must have its own challenge.

There are several things I need to address:

1. Using FAS to determine the material characteristics of each group is a great idea. But I think, analyzing the correlation between FAS categories  (low, average, high) with the associations of respondents regarding pregnancy, can be done and it would add more info to your paper.

2. It would be better if on table 1-8 if you add the proportion (%) of each items, so the readers can understand your data easier.

3. As for the results on the last tables (analysis of respondents' association and various parameters) we can see that there are several results of r (correlation coefficient) I think you can also add the strength of correlation , not only the significance between those parameters . I add the link to the classification: https://sphweb.bumc.bu.edu/otlt/MPH-Modules/PH717-QuantCore/PH717-Module9-Correlation-Regression/PH717-Module9-Correlation-Regression4.html

or perhaps it would be beneficial to consult with a statistician as well

Author Response

Reviewer 4

Thank you for the efforts you put in your research. Working with big data in such condition (COVID-19 pandemic) must have its own challenge.

There are several things I need to address:

  1. Using FAS to determine the material characteristics of each group is a great idea. But I think, analyzing the correlation between FAS categories  (low, average, high) with the associations of respondents regarding pregnancy, can be done and it would add more info to your paper.

No correlation was found between low, average, or high material status in FAS and the perception of pregnancy in women from Belarus. Similarly, no correlation was found between low, average, or high material status in FAS and the perception of pregnancy in women from Poland, Greece, Turkey, and Russia. (Results section)

  1. It would be better if on table 1-8 if you add the proportion (%) of each items, so the readers can understand your data easier.

Yes, however, there is a lot of data in the tables, and adding percentages will not have a positive effect on the readability of the tables.  

  1. As for the results on the last tables (analysis of respondents' association and various parameters) we can see that there are several results of r (correlation coefficient) I think you can also add the strength of correlation , not only the significance between those parameters . I add the link to the classification: https://sphweb.bumc.bu.edu/otlt/MPH-Modules/PH717-QuantCore/PH717-Module9-Correlation-Regression/PH717-Module9-Correlation-Regression4.html

We used the interpretations of the r values: poor correlation is r=0.1-0.2; fair r=0.3-0.5; moderate r=0.6-0.7; very strong r=0.8-0.9

  1. or perhaps it would be beneficial to consult with a statistician as well

We have cosult with a statistician

Round 2

Reviewer 2 Report

Comments and Suggestions for Authors

Dear Authors

Thank you for your time and effort. The changes implemented have improved the manuscript. Some of the proposed changes such as sample size estimation or the use of a validated fear of childbirth instrument have not been resolved, although this was difficult to resolve. I will accept your paper for publication, but you need to implement some additional changes to address these limitations:

-Include in the limitations section the non-performance of sampling calculation. The sample calculation should always be done; it is another question if the sample size calculated a priori cannot be obtained for different reasons (as in this case, for example, because of the pandemic).

-Include in the limitations a sentence warning about the low internal consistency of the instrument used (Cronbach's α of the scale Was 0.643).

-Indicate in which country the Bioethics Committee that approved this study is located.

Best regards

Author Response

Thank you for your comments.

-Include in the limitations section the non-performance of sampling calculation. The sample calculation should always be done; it is another question if the sample size calculated a priori cannot be obtained for different reasons (as in this case, for example, because of the pandemic).

In the study limitation, we have provided a sentence.

The sample size calculation has not been done.

-Include in the limitations a sentence warning about the low internal consistency of the instrument used (Cronbach's α of the scale Was 0.643).

In the study limitation, we have provided a sentence.

The Cronbach's α of the  FAS scale is 0.643 which indicates that it is not high.

-Indicate in which country the Bioethics Committee that approved this study is located.

The study was conducted with approval from the Bioethics Committee APK.002.587.2021, in Białystok, Poland.